# Influence of Farming Intensity and Climate on Lowland Stream Nitrogen

**Guillermo Goyenola** [1,*], **Daniel Graeber** [2], **Mariana Meerhoff** [1,3], **Erik Jeppesen** [3,4,5],
**Franco Teixeira-de Mello** [1], **Nicolás Vidal** [1], **Claudia Fosalba** [1], **Niels Bering Ovesen** [3],
**Joerg Gelbrecht** [6], **Néstor Mazzeo** [1,7] and **Brian Kronvang** [3]

1   Departamento de Ecología y Gestión Ambiental, Centro Universitario Regional Este (CURE), Universidad de la República, Punta del Este 20100, Uruguay
2   Department Aquatic Ecosystem Analysis, Helmholtz Centre for Environmental Research—UFZ, 39114 Magdeburg, Germany
3   Department of Bioscience and Arctic Research Centre, Aarhus University, 8600 Silkeborg, Denmark
4   Sino-Danish Centre for Education and Research, Beijing 100000, China
5   Limnology Laboratory, Department of Biological Sciences and EKOSAM, Middle East Technical University, 06800 Ankara, Turkey
6   Leibniz-Institute of Freshwater Ecology and Inland Fisheries, 12587 Berlin, Germany
7   South American Institute for Resilience and Sustainability Studies (SARAS), Bella Vista 20302, Uruguay
*   Correspondence: ggoyenola@cure.edu.uy or goyenola@gmail.com; Tel.: +598-9565-5636

**Abstract:** Nitrogen lost from agriculture has altered the geochemistry of the biosphere, with pronounced impacts on aquatic ecosystems. We aim to elucidate the patterns and driving factors behind the N fluxes in lowland stream ecosystems differing about land-use and climatic-hydrological conditions. The climate-hydrology areas represented humid cold temperate/stable discharge conditions, and humid subtropical climate/flashy conditions. Three complementary monitoring sampling characteristics were selected, including a total of 43 streams under contrasting farming intensities. Farming intensity determined total dissolved N (TDN), nitrate concentrations, and total N concentration and loss to streams, despite differences in soil and climatic-hydrological conditions between and within regions. However, ammonium ($NH_4^+$) and dissolved organic N concentrations did not show significant responses to the farming intensity or climatic/hydrological conditions. A high dissolved inorganic N to TDN ratio was associated with the temperate climate and high base flow conditions, but not with farming intensity. In the absence of a significant increase in farming N use efficiency (or the introduction of other palliative measures), the expected farming intensification would result in a stronger increase in $NO_3^-$, TDN, and TN concentrations as well as in rising flow-weighted concentrations and loss in temperate and subtropical streams, which will further exacerbate eutrophication.

**Keywords:** agricultural impact; stream; nitrogen concentration; nitrogen losses; eutrophication

## 1. Introduction

The changes promoted by agriculture in terrestrial and freshwater ecosystems can be quite dramatic [1,2]. During the 20th and 21st centuries, the global nitrogen (N) cycle has accelerated due to the artificial fixation of atmospheric $N_2$ (g) and extensive use of N fertilizers to boost agricultural production [3,4]. A high amount of reactive N (~100 Tg N·year$^{-1}$) is used in global agriculture, but it is estimated that less than 1 out of every 5 N-atoms used as fertilizer is finally consumed by humans [3]. The N lost from agricultural production ultimately alters the geochemistry of the biosphere and

particularly impacts aquatic ecosystems, thus contributing to eutrophication, degradation of water quality, and biodiversity loss [5–9].

At the catchment scale, the biogeochemical processes determining the natural fluxes of N from land to aquatic ecosystems mainly depend on climatic and hydrological regimes, and their interaction with local soil and geological conditions [10]. Consequently, stream N concentrations are dependent on variations in water temperature and discharge [11–13]. Moreover, farming intensity alters N mass balance and fluxes, leading to changes in the hydrochemistry of the streams [14]. One of the main changes induced by agriculture is the enhanced N level [4] due to the higher input of dissolved inorganic N (DIN), resulting in increased nitrate exports to aquatic ecosystems [15,16].

Intensive farming is a worldwide phenomenon [1,17]. Europe has a much longer history of intensive farming than tropics and subtropics, where the expansion of cropland areas became particularly rapid after 1850 [18]. Furthermore, 50% of the N fertilizer (inorganic and manure) used is confined to approximately 10% of the fertilized land, in tropical and subtropical areas of south and southeast Asia and southeast South America [19,20]. So far, most studies of the environmental consequences of N use in farming on streams have been conducted in developed countries, particularly in areas characterized by a temperate climate within Europe and North America [21], whereas only a few investigations have focused on developing countries and/or warm climate conditions e.g., [22–24]. Better knowledge of the use, transformation, and transport of N in subtropical and tropical climate regions—and thus of the N cycle—is urgently needed to generate more nutrient-efficient ways of producing food, while simultaneously reducing the negative side effects on the environment [18,25–27].

We aim to elucidate the patterns and driving factors behind the N fluxes in lowland stream ecosystems with contrasting land-uses and climate. Specifically, we aimed to analyze to what extent natural variations in soil characteristics and climate/hydrology would influence the effect of catchment farming on concentrations and N losses in lowland streams. We expected that streams draining microcatchments with high-intensity farming would have the highest concentrations of total N (TN), total dissolved N (TDN), nitrate ($NO_3^-$), ammonium ($NH_4^+$), and dissolved organic N (DON) and a higher DIN/TDN ratio than streams draining low-intensity farming microcatchments, independently of variations in edaphic variability or climatic-hydrological conditions.

## 2. Materials and Methods

To obtain a suitable balance between the different scales of analysis [28], the sampling strategy included three complementary monitoring schemes using different combinations of sampling frequencies and the associated number of streams (considered as replicates of farming intensity within each region).

A total of 43 lowland streams draining microcatchments under two contrasting conditions of agricultural intensity (hereafter farming intensity) were selected in two distinctive climate areas: humid cold temperate (Dfb sensu: [29]; n = 21 in summer; n = 20 in winter) and humid subtropical (Cfa sensu: [29]; n = 22; Table 1). The topography of both selected landscapes was characterized by gently rolling plains (mean slope < 5%) and the hydrographic catchment size varied around 10 km$^2$ (Denmark 9 ± 11 km$^2$, Uruguay 12 ± 7 km$^2$, average and standard deviation, respectively).

One stream per combination of farming intensity/climate-hydrology condition (n$_{streams}$ = 4, Table 1) was described in detail relative to hydrology, hydrochemistry, meteorological conditions, and catchment land-use. These streams acted as "benchmark streams" for each condition, and water samples with two different temporal resolutions were taken for a 2-year period (see Section 2.2). Both Danish benchmark streams are part of the Gudenå River basin, while the Uruguayan benchmark streams are part of the Santa Lucía Chico River basin. Besides the benchmark streams, a larger set of streams grouped according to farming intensity/climate-hydrology conditions were sampled with a "snapshot" approach (n$_{streams}$ = 39 winter, 38 summer; 1 sample per season; Table 1). More detailed information can be found in Goyenola et al. [30] and Graeber et al. [31]. Automated gauging stations were established in the four benchmark microcatchments. Hydrometric data were recorded every

10 minutes using CR10X data loggers (Campbell Scientific Ltd., Shepshed, UK). In the subtropical streams, we used CS450 Submersible Pressure Transducers (Campbell Scientific Ltd., Shepshed, UK) for water stage monitoring and Rain-O-Matic Professional rainfall automatized gauges (Pronamic). In temperate catchments, the water level was registered with PDCR 1830 pressure sensors (Druck), while meteorological information was obtained from the Danish Meteorological Institute monitoring the network based on a $10 \times 10$ km grid. Periodic instantaneous flow measurements were taken using a C2-OTT Kleinflügel, transferring data to software for the calculation of instantaneous discharge (VB-Vinge 3.0, Mølgaard Hydrometri).

**Table 1.** Characteristics of the soil and land-use of all the studied stream catchments according to the sampling method used. Benchmark streams were sampled fortnightly using grab and automatized pooled sampling.

| Climate & Farming Intensity | Benchmark Streams | Snapshot Grab-Sampling |
|---|---|---|
| TEMP Low | Granslev stream<br>Haplic Luvisols [#] O.M. = 5%<br>F.A.= 29%; mean L.U. = 0.25 ha$^{-1}$<br>N fertilizer = 45 kg N·ha$^{-1}$·year$^{-1}$<br>(45 % fertilizers; 55% manure) | Mainly Luvisols and Podsols; Arenosols [#]<br>O.M. < 5%. Range F.A. = 0%–26% |
| TEMP High | Gelbæk stream<br>Gleyic Luvisols [#] O.M. < 5%<br>F.A.= 92%; mean L.U. = 0.79 ha$^{-1}$<br>N fertilizer = 143 kg N·ha$^{-1}$·year$^{-1}$<br>(45 % fertilizers; 55% manure) | Mainly Luvisols and Podsols, some<br>Albeluvisols Arenosols and Cambisols [#]<br>O.M. < 5%. Range F.A. = 74%–93% |
| SUBT Low | Chal-Chal stream<br>Luvic Phaeozem and<br>Eutric Vertisols * O.M. = 5.2%<br>F.A.= 30%; mean L.U. = 0.62 ha$^{-1}$<br>N fertilizer = 76 kg N·ha$^{-1}$·year$^{-1}$<br>(18% fertilizers; 82% manure) | Phaeozem and Vertisols *<br>O.M. = 5% ± 1.5<br>Range F.A. = 0%–25% |
| SUBT High | Pintado Stream<br>Eutric Regosols * O.M. = 4% to 5%<br>F.A.= 90%; mean L.U. = 2.00 ha$^{-1}$<br>N fertilizer = 242 kg N·ha$^{-1}$·year$^{-1}$<br>(17% fertilizers; 83% manure) | Mainly Phaeozem *<br>O.M. = 5% ± 1.5<br>Range F.A. = 75%–100% |
| Total n | $n_{streams}$ = 4 | $n_{streams}$ = 39 (w), 38 (s) |
| Sampling | grab and pooled sampling<br>2 years | 1 sample in (w) and 1 in (s) |

Abbreviations: TEMP: temperate streams; SUBT: subtropical streams; Low and High: low and high-farming intensity; O.M.: organic content of soils (%); F.A.: percentage of the Farming area; mean L.U.: mean livestock units by ha; N fertilizer: total N inputs by year, discriminated in contributions of fertilizers and manure. (w): winter; (s): summer. Source: (#) World Reference Soil Database classification, European Commission and European Soil Bureau Network (2004); (*) SOTERLAC database, ISRIC Foundation (www.isric.org).

Nitrogen catchment input by hectare and year in Danish catchments was estimated considering the surface of agriculture land in the catchment multiplied by the national average of chemical fertilizer use for the years 2011–2012 (total input 69 kg N·ha$^{-1}$·year$^{-1}$), and the livestock density multiplied by the average of N production in manure by livestock (86 kg N·ha$^{-1}$·year$^{-1}$ [32]). Nitrogen catchment input by hectare and year in Uruguayan catchments was estimated through interviews with the technical managers of the establishments. Based on the best available knowledge considering empirical data [33], the average of N production in manure for the Uruguayan was assumed to equal than for Danish catchments. The total N input was divided by the total catchment area to be able to make quantitative comparisons with the TN losses.

## 2.1. Farming Intensity

Land-use intensity is a complex multidimensional concept and is difficult to measure [34,35], and therefore we applied explicit operational definitions (Table 1). Low-intensity farming catchments represent the condition with minimal anthropogenic pressures for each region. The subtropical low-intensity farming catchments (n = 9, both summer and winter) were dominated by the natural

grasslands of the Pampa Biome [36] and sustained low-density cattle production (below one head per hectare), while a mixture of deciduous and coniferous forests dominated the temperate low-intensity farming catchments ($n_{summer}$ = 8; $n_{winter}$ = 9; Figure 1; Table 1). Less than 30% of the selected low-intensity farming catchments in both countries were influenced by arable cropping systems.

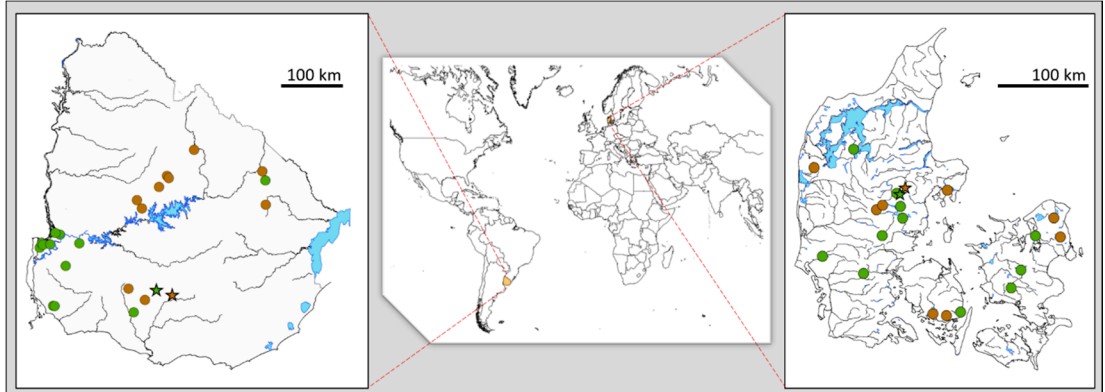

**Figure 1.** The geographic location of sampled streams. Left: subtropical/Uruguayan streams. Right: temperate/Danish streams. Circles: Snapshot grab-sampling. Stars: Benchmark streams. Green: high-farming intensity; brown: low-farming intensity. LAT/LONG of benchmark streams: TEMP Low: 56.2837/9.8975; TEMP High: 56.2254/9.8117; SUBT Low: −33.8256/−56.2821; SUBT Low: −33.9036/−56.0064. All the catchments fall within polygons limited by the following coordinates: Denmark 55.09 to 56.64 N; 8.38 to 12.44 E, Uruguay −31.79 to −34.18; −54.41 to −58.31 (decimal degrees; WGS84).

The criteria for the selection of high-intensity farming catchments were: (1) that arable cropping systems with intensive use of fertilizers affected more than 70% of the total area and (2) that they represented real and typical high-intensity farming catchments in the climatic-hydrological regions of the two countries ($n_{TEMP}$ = 12; $n_{SUBT}$ = 13; Table 1).

*2.2. Hydrochemistry Monitoring*

The complementary strategy allowed us to evaluate whether the patterns observed were generalizable for each climate/hydrological region or even common to both. The monitoring approach included:

- Fortnightly grab-sampling in benchmark streams: Sub-surface grab samples were taken from a well-mixed section with no macrophytes in the center of the stream channel during the daytime. This instantaneous sampling was used for the analysis of conservative and non-conservative N fractions (i.e., TN, TDN, $NO_3^-$, DON, and $NH_4^+$).
- Automatic pooled sampling in benchmark streams: High-frequency monitoring using automated equipment was conducted during the same two-year period. Glacier refrigerated automatic samplers (ISCO-Teledyne) collected an equal water volume every four hours from the same sampling point, and the pooled samples were collected fortnightly. The final nutrient concentration in the only sampler carboy thus represented a time-proportional average for the fortnightly sampling period. As this sampling involved refrigerated storage of pooled samples for up to two weeks, the emphasis was placed on the analysis of TN.
- Snapshot grab-sampling in the series of streams was made once in winter and once in summer. Sub-surface grab samples were taken in a well-mixed section with no macrophytes from the center of the stream channels during the daytime. This instantaneous sampling was used for the analysis of different N fractions, with emphasis on dissolved compounds (i.e., TDN, $NO_3^-$, DON, and $NH_4^+$).

## 2.3. Laboratory Measurements

All pooled water samples from the high frequency monitored streams were analyzed for total N (TN), total dissolved N (TDN), and nitrate ($NO_3^-$). Analysis of fortnightly and snapshot samples included TN, TDN, $NO_3^-$, $NH_4^+$, and DON (dissolved organic N). Different techniques were applied to guarantee accuracy, address the different concentration ranges, and assure the inter-comparability of results between countries.

Water samples for the determination of dissolved N fractions were filtered through 0.45-μm membrane filters pre-rinsed with ultrapure water (Milli-Q water). TN was converted to nitrate following the protocol of Valderrama [37] and analyzed as $NO_3^-$. In Uruguay, the standard sodium salicylate method was used for $NO_3^-$ determination [38,39]. For the Danish samples, the sum of both nitrate and nitrite was determined by flow analysis (CFA and FIA) and spectrometric detection ([40], Danish Standard 223). Additionally, the samples were analyzed by segmented flow analysis including an additional channel to measure $NO_2^-$. As the $NO_2^-$ concentration was always below the quantification limit of the technique ($<0.01$ mg N·L$^{-1}$), it was not considered in the analysis; instead we assumed that $(NO_3^-) + (NO_2^-) = (NO_3^-)$.

The analysis of the Uruguayan samples used for total TDN determination followed the same approach as for the TN samples. The TDN concentrations of the Danish samples were measured using high-temperature catalytic oxidation (HTCO, multi N/C 3100, Jena Analytik, Jena, Germany) after acidifying the samples to pH 2–3 with HCl and sparging with synthetic air for 5 min. The samples were oxidized with a platinum catalyst at 700 °C in a synthetic air stream, and TDN was measured as NO gas with a chemiluminescence detector [41]. For the Danish water samples with high $NO_3^-$ levels (from high-intensity farming), the HTCO method led to significant underestimation of TDN, likely because the HTCO method did not permit oxidization of all the N [42,43]. When oxidation was not possible, TDN was estimated as the addition of DIN + DON, DIN being in turn estimated as the addition of $NO_3^-$ and $NH_4^+$ and DON being measured by size-exclusion chromatography [43,44]. DON samples taken in subtropical streams were acidified with hydrochloric acid and frozen, following Hudson, et al. [45], and sent for analysis at the Leibniz-Institute of Freshwater Ecology and Inland Fisheries laboratory in Berlin. Before measurements, the samples were brought to the same target pH level of 7.5 ± 0.5 by neutralization with sodium hydroxide. $NH_4^+$ was measured following the indophenol-blue method [46].

## 2.4. Data Analysis

Non-linear regressions between stage and discharge at each monitoring station (rating curves) were fitted. Rating curves were used to generate a discharge data series with a 10-min resolution using the software HYMER (www.orbicon.com). Base flow index (BFI) was estimated for the complete data set from daily hydrographs using the automatic routine proposed by Arnold et al. [47] to set the magnitude of the groundwater contribution to the streamflow.

Total N concentrations were determined from the fortnightly samples, while the high frequency automatized pooled data were used to estimate the annual TN transport, loss, and flow-weighted TN concentrations in the subset of the four benchmark streams (2-year period). The TN transport was calculated by multiplying the TN concentration obtained from the pooled samples by the accumulated discharge for the same fortnightly period and summing yearly [48]. The TN loss was calculated dividing the annual transport by the catchment area in hectares [49]. Missing data from the relatively short periods when the automatic samplers were not in operation (e.g., due to freezing in Denmark) were re-generated through linear interpolation of concentrations [50]. The flow-weighted concentration (FWC) was calculated as the annual TN transport divided by the annual runoff. Dissolved N fractions were analyzed from both the fortnightly grab samples in the four benchmark streams (2-year period) and the snapshot samples ($n_{streams}$ = 39 winter, 38 summer; 1 sample per season).

The factorial design relative to climate/hydrology conditions and farming intensity was evaluated using two-way nested ANOVA with farming intensity nested within climate/hydrology, followed

by a post hoc pairwise multiple comparison when appropriate [51]. Variability in the high frequency-automatized pooled and fortnightly samples represents the temporal variation within each intensively sampled stream, while variability in the snapshot sampling expresses spatial variation among comparable systems. The relationship between TN concentrations from the fortnightly instantaneous grab samples, discharge, and water temperature were analyzed by Spearman rank-order correlations.

## 3. Results

### 3.1. Climate and Hydrology

The climatic characteristics in the study period (2010 to early 2012) can be considered typical for both Denmark and Uruguay (Table 2; [30]). The annual average air temperature did not exhibit any anomaly [52,53] and corresponded to the mean for the corresponding region recorded by national meteorological services based on recent historical information [54,55]. Mean air temperature was 8.8 °C and ranged between −7.0 and 20.4 °C in the temperate sites and was ca. 17.5 °C, ranging between 3.7 to 32.2 °C, in the subtropical sites. No dry or wet seasons occurred in either country, but marked differences in frequency and intensity of rainfall were detected. Total annual precipitation was lower in the Danish catchments than in the Uruguayan catchments (Table 2), while rain events were less intense but more frequent in the Danish than in the Uruguayan catchments [30,31]. Thus, hydrologically, the Danish streams are more stable than the Uruguayan catchments, the latter being described as "flashy" in previous publications (Richards-Baker Flashinnes Index < 0.3 for Danish streams and > 0.9 for comparable Uruguayan streams; [30]). The Danish stable streams have much higher contribution of groundwater to water flow (higher base flow index, Table 2).

**Table 2.** Main climatic and hydrological characteristics of the four benchmark catchments monitored at high frequency (nstreams = 4, 2-year period), showing annual accumulated rainfall in mm for each study year (sources: [a] [56], [b] [55]). Abbreviations: TEMP: temperate streams; SUBT: subtropical streams; Low and High: low and high-farming intensity.

| Characteristic | TEMP Low | TEMP High | SUBT Low | SUBT High |
|---|---|---|---|---|
| Accumulated rainfall of each study year (mm·y$^{-1}$) | 756–770 | 766–778 | 1010–1030 | 1196–1405 |
| Mean regional accumulated rainfall (mm·y$^{-1}$) | 765 [a] | | 1100–1200 [b] | |
| Base Flow Index (BFI) | 0.88 | 0.64 | 0.39 | 0.29 |

### 3.2. Total Nitrogen Concentrations and Losses in Benchmark Streams

The farming intensity was, as expected, a strong determinant factor of stream TN concentrations, while climate/hydrology had no significant effects on the TN concentrations in the benchmark streams (Figure 2). This was expressed by low and not statistically different TN concentrations in the low-farming intensity streams (varying around 1.0 mg N·L$^{-1}$) and significantly higher average annual TN concentrations in high-intensity farming streams (2.2 ± 1.4 mg N·L$^{-1}$ and 4.3 ± 2.5 mg N·L$^{-1}$ for the subtropical and temperate high-intensity farming streams, respectively; mean ± SD; Figure 2). Total N loss and flow-weighted concentrations of TN (TN-FWC) obtained through high frequency-automatized pooled sampling were also higher in the highest intensity farming streams for both climatic/hydrological conditions (annual estimations for two years, no statistical testing possible; Table 3).

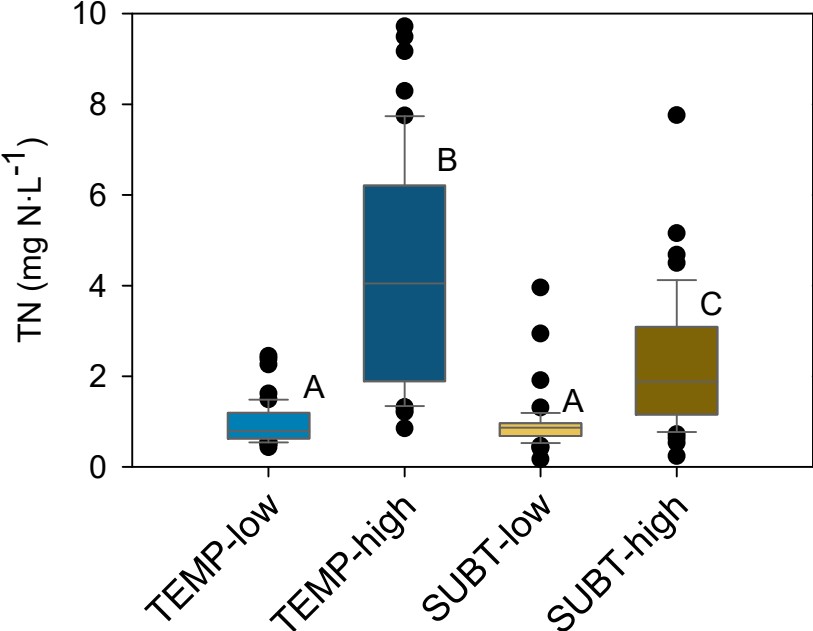

**Figure 2.** Variability in total nitrogen (TN) concentrations for each of the four fortnightly grabs sampled benchmark streams. Data correspond to a 2-year period. ANOVA results: F TEMP vs. SUBT (1,190) = 23.73, Farming intensity conditions nested in climate/hydrology (2,190) = 70.44, F Interaction (1,190) = 378.9. P < 0.001 for all cases. A, B, and C describe statistically similar groups according to Bonferroni post hoc tests. The upper and lower boundaries of the box mark the 25th and 75th percentile, whiskers above and below the box indicate the 90th and 10th percentiles and the line within the box marks the median. Black dots display outliers. Abbreviations: TEMP: temperate streams; SUBT: subtropical streams; low and high: low and high-farming intensity in the catchments.

**Table 3.** Total nitrogen losses by hectare (kg N·ha$^{-1}$·year$^{-1}$) and flow-weighted concentrations (FWC; mg N·L$^{-1}$) estimated annually using high frequency automatized pooled sampling of the four benchmark streams (water samples taken every 4 h and accumulated fortnightly). Abbreviations: TEMP: temperate streams; SUBT: subtropical streams; low and high: low and high-farming intensity in the catchments.

| Region | Year | Low-Farming Intensity | | High-Farming Intensity | |
|---|---|---|---|---|---|
| | | TN Loss | FWC TN | TN Loss | FWC TN |
| SUBT | 1 | 1.39 | 0.82 | 4.67 | 1.99 |
| SUBT | 2 | 2.12 | 0.72 | 9.17 | 2.13 |
| TEMP | 1 | 6.11 | 1.2 | 13.16 | 6.28 |
| TEMP | 2 | 4.68 | 0.98 | 12.65 | 6.23 |

Without trends associated with farming intensity, the total N lost from subtropical catchments was between 2% and 4% of the total annual inputs as fertilizers and manure (Tables 1 and 3). For the case of temperate catchments, the total N lost by the stream was 9% for high-intensity farming streams and between 10% and 14% (year 1 and 2 of monitoring) in low-intensity farming streams (Tables 1 and 3).

*3.3. Influence of Temperature and Discharge on Total Nitrogen Concentrations*

Stream TN concentrations tended to decrease with increasing water temperature and decreasing discharge, as reflected in the set of four fortnightly sampled benchmark streams (Figure 3). In both climates, the relationship between TN concentrations and discharge showed a higher explained variance for high-intensity farming than for low-intensity farming (Figure 3). The TN concentration of subtropical low-intensity farming stream, however, did not exhibit statistical relationships with temperature and discharge (Figure 3).

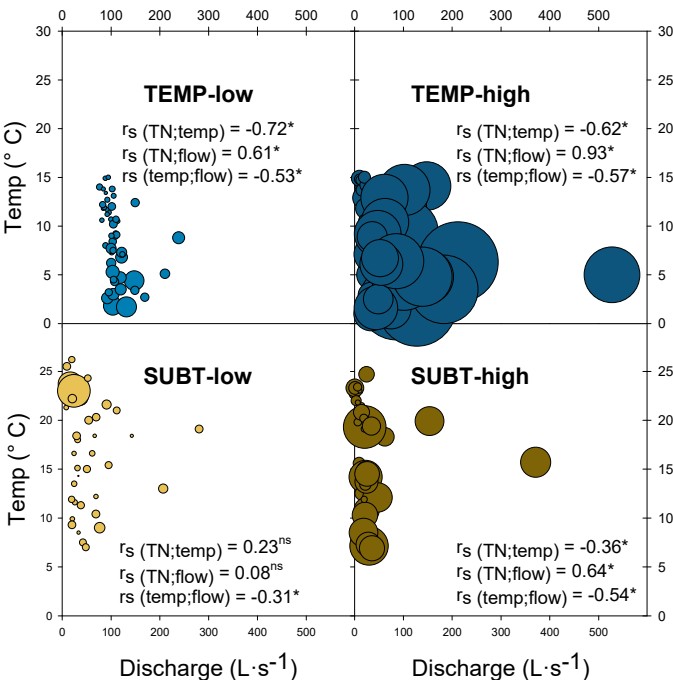

**Figure 3.** Total nitrogen concentrations vs. water temperature and discharge in the subset of the four benchmark streams under high-frequency monitoring. Bubble size represents the concentration of TN from fortnightly grab sampling (two-year data). The graphs show the main environmental gradients for each stream. Spearman rank-order correlations (rs) are marked with * when significant ($0.01 \leq p \leq 0.05$). ns: non-significant. Abbreviations: TEMP: temperate streams; SUBT: subtropical streams; low and high: low and high-farming intensity in the catchments.

*3.4. Influence of Climate/Hydrology and Farming Intensity on Nitrogen Species*

Total dissolved N (TDN) constituted the main fraction of TN in all the studied streams (Figures 2 and 4). It was also affected by farming intensity regardless of climate/hydrological conditions and monitoring method/sampling time (Table 4; Figure 4). These assertions are based on the comparable results obtained in all the sampled streams (n = 43), including both monitoring schemes: fortnightly grab sampling in benchmark streams and snapshot sampling (the analyses of this section include all these data sets; see Figure 4).

Relatively low (range of averages: 0.4 to 0.9 mg N·L$^{-1}$), and not statistically different average TDN concentrations were found in all low-intensity farming streams, intermediate concentrations were found in the subtropical-high intensity farming streams (1.2 to 1.8 mg N·L$^{-1}$), and the highest concentrations occurred in the temperate high-intensity farming streams (3.5 to 5.2 mg N·L$^{-1}$; Table 4; Figure 4 upper panels). The pattern of intermediate TDN concentrations in the subtropical high-intensity farming streams was similar for the two sampling strategies, the only significant difference with low-intensity farming streams occurring for the fortnightly samples (Table 4; Figure 4).

Nitrate (NO$_3^-$) concentrations resembled the above TN and TDN: low and not statistically different NO$_3^-$ concentrations in the low-intensity farming streams, intermediate in the subtropical high-intensity farming streams, and the highest in the temperate high-intensity farming streams (complete data set; Table 4; Figure 4). Average annual NO$_3^-$ concentrations ranged between values as low as 0.05 and 0.3 mg N·L$^{-1}$ in the low-intensity farming streams in both climates (Figure 4). Average nitrate concentrations in the subtropical high-intensity farming streams ranged between 0.4 and 0.8 mg N·L$^{-1}$. In contrast, in the temperate high-intensity farming streams, nitrate varied between 3.2 and 4.9 mg N·L$^{-1}$ on average, depending on the sampling method and season (Figure 4).

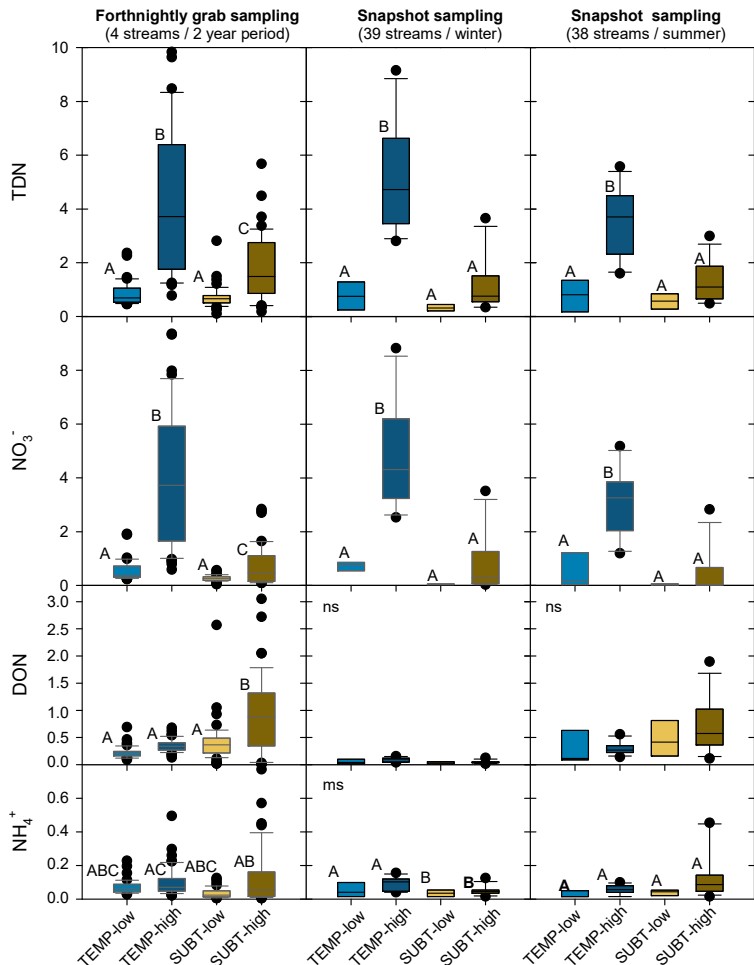

**Figure 4.** The concentration of dissolved nitrogen fractions in fortnightly grab sampling in benchmark streams and snapshot samples. Significance level P < 0.05. A, B, and C describe statistical groups according to post hoc Bonferroni tests. We indicate non-significant results as ns (p > 0.1), and marginally significant as ms (0.05 < P < 0.1). Abbreviations: TEMP: temperate streams; SUBT: subtropical streams; low- and high-FI: low and high-farming intensity. Vertical axes are concentrations expressed in mg N·L$^{-1}$. The upper and lower boundaries of the box mark the 25th and 75th percentile, whiskers above and below the box indicate the 90th and 10th percentiles and the line within the box marks the median. Black dots display outliers. Note that the scale varies among fractions.

We found no significant association between DON concentrations and climate/hydrological conditions or farming intensity, except for significantly higher levels of DON in the subtropical high-intensity farming streams (benchmark streams only, average = 0.9 mg N·L$^{-1}$; Figure 4). All the other streams representing both climatic conditions had average levels below 0.7 mg N·L$^{-1}$ (Table 4; Figure 4).

No significant relationships were found between ammonium (NH$_4^+$) concentrations and climate/hydrological conditions and farming intensity (fortnightly grab sampling in benchmark streams and snapshot sampling in summer; Table 4; Figure 4). Average NH$_4^+$ concentrations were always < 0.1 mg N·L$^{-1}$, regardless of climate and farming intensity. In the wintry snapshot sampling, average NH$_4^+$ concentrations were significantly higher in the temperate (average 0.08 mg N·L$^{-1}$) than in the subtropical streams (0.04 mg N·L$^{-1}$; Table 4; Figure 4).

The DIN/TDN ratio was higher in the temperate (average ranging from 0.5 to almost 1) than in the subtropical streams (average ranging from 0.2 to 0.5) for the fortnightly and snapshot sampled streams (Table 4; Figure 5). The temperate high-intensity farming streams exhibited the highest DIN/TDN ratios (average between 0.89 and 0.95), which was linked to the strong predominance of

$NO_3^-$ (Figures 4 and 5). In general, higher variability in the DIN/TDN ratio was observed in the subtropical streams (SD ranging from 0.2 to 0.3; Figure 5) than in the temperate streams.

**Table 4.** Summary of 2-way nested ANOVA tests for the concentration of N forms, indicating the origin (sampling method) of the data. Above: Main effects of climate/hydrology conditions (two levels). Below: Main effects of farming intensity (two levels: low- and high-intensity farming) nested within climate/hydrology conditions and interaction between factors. F values and the respective degrees of freedom are indicated. Significance level: $P > 0.1$ ns, $0.05 < P < 0.1$ ms, $P < 0.05$ *, $P < 0.01$ **, $P < 0.001$ ***. Results of post hoc pairwise multiple comparisons are shown in Figures 3–5. Abbreviations: TEMP: temperate streams; SUBT: subtropical streams; low and high: low- and high-farming intensity in the catchments.

| N Form | Benchmark Streams | Snapshot Sampling | |
| --- | --- | --- | --- |
| | Fortnightly Sampling | Winter | Summer |
| | Comparison between Climate/Hydrology Conditions (TEMP vs. SUBT) | | |
| TDN | $F(1, 181) = 38.16$ *** | $F(1, 35) = 29.97$ *** | $F(1, 34) = 19.60$ *** |
| $NO_3^-$ | $F(1, 186) = 78.25$ *** | $F(1, 35) = 29.99$ *** | $F(1, 34) = 30.58$ *** |
| DON | $F(1, 181) = 27.27$ *** | $F(1, 35) = 1.44, p = 0.24$ ns | $F(1, 34) = 4.90$ * |
| $NH_4^+$ | $F(1, 186) = 0.19, p = 0.66$ ns | $F(1\ 35) = 7.72$ ** | $F(1, 34) = 2.31, p = 0.14$ ns |
| DIN/TDN | $F(1, 181) = 179.67$ *** | $F(1, 35) = 30.06$ *** | $F(1, 34) = 23.67$ *** |
| | Comparison between Farming Intensity Conditions (Low & High) Nested in Climate/Hydrology | | |
| TDN | $F(2, 181) = 71.85$ *** | $F(2, 35) = 30.30$ *** | $F(2, 34) = 20.78$ *** |
| $NO_3^-$ | $F(2, 186) = 80.12$ *** | $F(2, 35) = 28.48$ *** | $F(2, 34) = 20.49$ *** |
| DON | $F(2, 181) = 24.42$ *** | $F(2, 35) = 0.232, p = 0.79$ ns | $F(2, 34) = 1.19, p = 0.31$ ns |
| $NH_4^+$ | $F(2, 186) = 10.05$ *** | $F(2, 35) = 2.75, p = 0.08$ ms | $F(2, 34) = 3.09, p = 0.08$ ms |
| DIN/TDN | $F(2, 181) = 14.67$ *** | $F(2, 35) = 6.26$ ** | $F(2, 34) = 5.50$** |
| | Interaction between Farming Intensity and Climate/Hydrology | | |
| TDN | $F(1, 181) = 307.1$ *** | $F(1, 35) = 85.0$ *** | $F(1, 34) = 116.4$ *** |
| $NO_3^-$ | $F(1, 186) = 193.6$ *** | $F(1, 35) = 57.5$ *** | $F(1, 34) = 52.33$ *** |
| DON | $F(1, 181) = 255.1$ *** | $F(1\ 35) = 103.0$ *** | $F(1, 34) = 59.9$ *** |
| $NH_4^+$ | $F(1, 186) = 140.3$ *** | $F(1, 35) = 106.0$ *** | $F(1, 34) = 21.6$ *** |
| DIN/TDN | $F(1, 181) = 2744.1$ *** | $F(1, 35) = 299.3$ *** | $F(1, 34) = 132.9$*** |

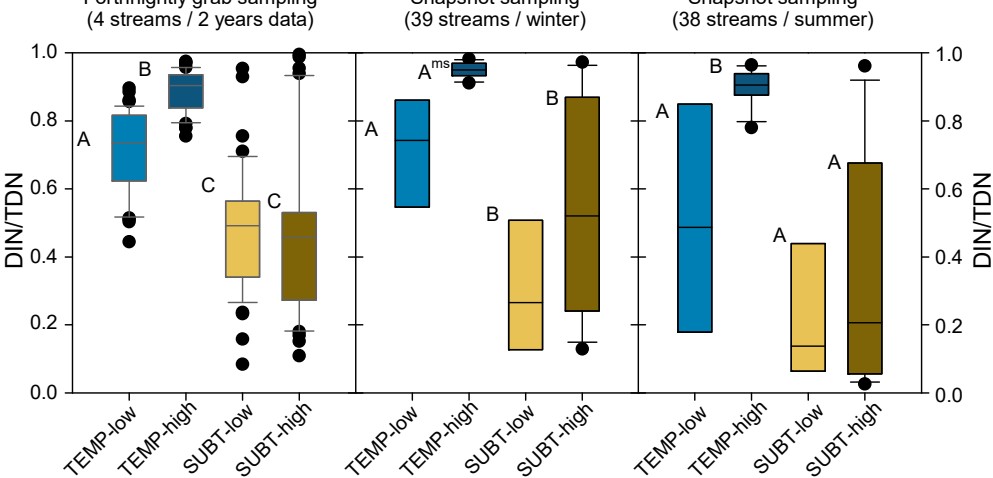

**Figure 5.** DIN/TDN ratio for the fortnightly grab sampling in benchmark streams and snapshot samples. Significance level $P < 0.05$. A, B, and C describe statistical groups according to post hoc Bonferroni tests. We indicate marginally significant as ms ($0.05 < P < 0.1$). The upper and lower boundaries of the box mark the 25th and 75th percentile, whiskers above and below the box indicate the 90th and 10th percentiles and the line within the box marks the median. Black dots display outliers. Note that the scale varies among fractions. Abbreviations: TEMP: temperate streams; SUBT: subtropical streams; low and high: low and high-farming intensity.

## 4. Discussion

### 4.1. Influence of Farming Intensity

Our analysis of streams draining microcatchments under low-intensity farming conditions in contrasting climatic-hydrological settings revealed low and quite comparable TN, TDN, $NO_3^-$, $NH_4^+$, and DON concentrations. In addition, $NO_3^-$ concentrations in low-intensity farming streams at both climate/hydrological conditions exhibited levels that were considered as background concentrations in a recent and independent study conducted for streams draining relatively undisturbed catchments in Denmark and elsewhere [57]. No reference data for background concentrations in Uruguayan or subtropical streams have previously been reported in the scientific literature.

In contrast, the highest concentrations of $NO_3^-$ were found in all sampled streams draining microcatchments impacted by high-intensity farming, irrespective of the monitoring method. Annual flow-weighted concentrations of TN never exceeded 1.2 mg $N \cdot L^{-1}$ in the two benchmark streams draining low intensity farmed catchments, but they were always higher than 2.0 mg $N \cdot L^{-1}$ in the two benchmark streams draining high intensity farmed catchments. The higher $NO_3^-$ concentrations in the water (leading to higher concentrations of TDN and TN) can, therefore, be attributed to the impact of intensive farming in the catchments.

The TN, TDN, $NO_3^-$, and TN-FWC concentrations in the streams draining high-intensity farming catchments were significantly higher in the temperate climate, with stable discharge conditions than in the subtropical climate with flashy discharge conditions.

The N input to the catchments as fertilizer and manure, was higher in subtropical than in temperate catchments, particularly by the higher contribution of manure derived from higher livestock loads. Contrarily, the N loss/N input fraction was higher in temperate catchments (9% to 14%), respect to subtropical ones (2% to 4%). Further, more detailed studies must be done to establish if assumptions made about N content of manure for Uruguay are correct, or if our results could be biased by it. Notwithstanding, these results are consistent with the much longer history of intensive farming in central and northern Europe than in South America, creating a potentially high N legacy in the groundwater feeding the streams [58–60]. Moreover, the widespread use of artificial drainage practices via tile drains in Danish productive catchments is a shortcut pathway for nitrate from the soils to surface waters; thus avoiding attenuation processes in groundwater [61–64]. Accordingly, the streams draining the temperate high intensity farmed catchments were characterized by higher $NO_3^-$ concentrations and a larger contribution of groundwater to the total flow measured (higher base flow index, BFI) than in similar subtropical streams.

The concentrations of $NH_4^+$ and DON in streams did not show any clear relationship with the analyzed environmental factors. The proportional contribution of DON to TDN in the subtropical streams was, however, higher than in the temperate streams, which might be explained by the moderate to low levels of $NO_3^-$ observed in the subtropical streams.

The global use of N fertilizers increases steadily [65], and the trend is forecasted to continue for the next decades despite more efficient management practices [66]. In the absence of a significant increase in N use efficiency (or the introduction of other retention or mitigation measures), the expected farming intensification in the future will result in an increase in N concentrations and losses in streams, which will further exacerbate eutrophication in surface freshwater bodies.

### 4.2. Influence of Climate

Benchmark streams showed statistically significant relationships between N concentrations and water temperature (negative) and discharge (positive). The former may be linked with high temperature-driven denitrification and higher N assimilation by aquatic macrophytes and periphyton in summer [65–67]. The latter, in contrast, may be explained by the N legacy in groundwaters in the intensively farmed catchments together with the annual N surplus and hence diffuse N contributions from agriculture in the catchments (leading to higher $NO_3^-$ and TN concentrations in streams with

increasing discharges). The lack of a significant relationship between N concentrations and temperature and discharge in the subtropical stream draining the low intensity farmed catchment was probably caused by the extremely low or lack of N surplus and N legacy, together with high denitrification and biological N uptake promoted by the higher temperatures.

Our results suggest that in a stationary scenario of farming intensity and management, warming alone might promote a reduction of TN concentrations in lowland low-order streams driven mainly by a reduction in $NO_3^-$ concentrations. In contrast, the predicted increase in annual precipitation and the enhanced intensity of precipitation events in both countries [67–69] will probably increase the risk of diffuse N losses to streams, at least in intensively farmed catchments. Consequently, given these contradictory trends, it is uncertain what will be the resulting impact of climate change on N concentrations and losses in lowland streams in different climates and agricultural systems [70]. Several model scenario studies of climate change effects on N cycling in catchments have, however, suggested increases in exported N in the temperate climate regions [71,72]. The sense of the changes that the water tables suffer (e.g., height, residence time) probably will be one of the most influential factors regarding the N loss towards the streams. If the increase in flow regime variability, flashiness, and enhanced evapotranspiration results in a decrease in the contribution of groundwater [12], a lowering of the $NO_3^-$ concentration and DIN/TDN ratio could be expected. Nevertheless, if the increasing precipitations get more infiltration, higher groundwater tables, and longer periods with tile drain flow, the effects could be the contrary.

## 5. Conclusions

The results from our three complementary monitoring approaches were broadly comparable and support that farming intensity is of key importance for determining N concentrations and losses in lowland streams, despite differences in soil and climatic-hydrological conditions between and within regions.

Overall, farming intensity determines the concentrations of TN, TDN, and $NO_3^-$, flow-weighted TN, and TN exported to streams, but not those of ammonium ($NH_4^+$) and dissolved organic N (DON).

In the absence of a significant increase in farming N use efficiency (or the introduction of other palliative measures), the expected farming intensification will result in a stronger increase in $NO_3^-$, TDN, and TN concentrations as well as rising flow-weighted N concentrations and N losses in temperate and subtropical streams, further exacerbating eutrophication.

In contrast to our expectations, a high dissolved inorganic N (DIN) to TDN ratio was associated with temperate climate and high base flow conditions but not with farming intensity.

The consequences of changes in climate for the streams in the studied countries are hard to predict as higher temperatures and higher precipitation had contrasting effects on TN concentrations in our study.

**Author Contributions:** Conceptualization, B.K., E.J., M.M., and G.G.; data curation, G.G. and D.G.; formal analysis, G.G.; funding acquisition, B.K., E.J., and M.M.; investigation, all authors; project administration, M.M.; supervision, B.K., E.J., and M.M.; visualization, G.G.; writing—original draft preparation, G.G.; writing—review and editing, D.G., M.M., E.J., F.T.-d.M., N.V., N.M., and B.K. All authors have read and agreed to the published version of the manuscript.

**Funding:** The study was funded by the Danish Council for Independent Research, a grant by ANII-FCE (2009-2749) Uruguay, and the National L'Oreal-UNESCO Award for Women in Science-Uruguay with support from DICyT granted to MM. GG, FTM, MM, IGB, and NM received support from the SNI (Agencia Nacional de Investigación e Inovación, ANII, Uruguay). GG was supported by a Ph.D. scholarship from PEDECIBA. E.J. and D.G. were further supported by MARS (Managing Aquatic ecosystems and water Resources under multiple Stress) funded under the 7th EU Framework Programme, Theme 6 (Environment including Climate Change), Contract No.: 603378 (http://www.mars-project.eu). DG was also supported by a grant from the Danish Centre for Environment and Energy.

**Acknowledgments:** We thank the technicians of AU in Silkeborg for their work in the field and lab. In Uruguay, we are grateful for the field assistance from Iván González-Bergonzoni, Natalie Corrales, Anahí López-Rodríguez, Clementina Calvo, Gonzalo Moreno, Alfonsina López, Carlos Iglesias, Juan Clemente, Mariana Vianna, and Juan Pablo Pacheco, and also to Verónica Ciganda, Diego Michelini and Carlos Perdomo for their

generosity sharing agronomic information. We thank especially Daniella Agrati and family and the landowners (Mendiverri & Laturre) in Uruguay.

**Conflicts of Interest:** The authors declare no conflict of interest. The funders had no role in the design of the study; in the collection, analyses, or interpretation of data; in the writing of the manuscript, or in the decision to publish the results.

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
