# Peer review of "Influence of Farming Intensity and Climate on Lowland Stream Nitrogen"

_water, doi:10.3390/w12041021_

Round 1

Reviewer 1 Report

That manuscript describes background concentrations of different N forms in temperate and subtropical streams, as well as influence of climate and human activity (agriculture) on those. Especially reported background concentrations are commonly found rare but needed in several administrative contexts. The study was sound and carefully wrtten.

I was still missing some aspects:

1) Introduction: History of agriculture in Europe. That was one explanation in Discussion, but there is nothing of that in Introduction.

2) Materials and methods: What is the organic matter content of different soil types? Maybe that is one explanation for differences in TON concentration?

3) Materials and methods: How well the N content in manure is comparable? What is the main difference in animal husbandry, and their diets?

4) Is it possible to say anything of the eutrophication potential in different sttreams?

Reviewer 2 Report

Respected Authors,

 The paper is technically very sound and well written. Can the authors please address the 7 detailed comments mentioned in the attached pdf file. 

Reviewer 3 Report

This manuscript presents a study to evaluate influence of farming and climate on lowland stream nitrogen. The manuscript is generally well written with data and results clearly described and discussed. This study may help expand knowledge base in water quality study area, due to its focus on agricultural and climate impact on nitrogen losses and concentration as well as eutrophication. So in this sense I endorse the publication of this manuscript. My only suggestion on improving the manuscript is to add a chapter regarding studied areas (after Introduction) with data regarding this area and with maps of studied streams with marked water sampling locations. Other parts of the manuscript seem OK. I am looking forward to see this interesting paper published after performing of recommended minor revision.

Round 2

Reviewer 1 Report

I find the corrections to the manuscript adequate, and I understand that lach of observations set limits to the manuscript.